# Evaluation of the Mechanical and Biocidal Properties of Lapacho from Tabebuia Plant as a Biocomposite Material

**DOI:** 10.3390/ma14154241

**Published:** 2021-07-29

**Authors:** Magdalena Stepczyńska, Alona Pawłowska, Krzysztof Moraczewski, Piotr Rytlewski, Andrzej Trafarski, Daria Olkiewicz, Maciej Walczak

**Affiliations:** 1Department of Materials Engineering, Kazimierz Wielki University, 85-064 Bydgoszcz, Poland; alona.pawlowska@ukw.edu.pl (A.P.); kmm@ukw.edu.pl (K.M.); prytlewski@ukw.edu.pl (P.R.); trafarski@ukw.edu.pl (A.T.); 2Department of Environmental Microbiology and Biotechnology, Nicolaus Copernicus University in Toruń, 87-100 Toruń, Poland; deo@doktorant.umk.pl (D.O.); walczak@umk.pl (M.W.)

**Keywords:** biocomposites, biodegradable materials, filler, cortex fibers, Lapacho, Tabebuia plant

## Abstract

The aim of this article is to discuss in detail the physicochemical properties of polylactide (PLA) reinforced by cortex fibers, which may cause bacterial mortality and increased biodegradation rates. PLA biocomposites containing cortex Lapacho fibers from Tabebuia (1–10 wt%) were prepared by extrusion and injection moulding processes. The effects of Lapacho on the mechanical and biocidal properties of the biocomposites were studied using tensile and impact tests, dynamic mechanical analysis (DMA), differential scanning calorimetry (DSC), thermogravimetry (TG), and the method of evaluating the antibacterial activity of antibacterial treated according to the standard ISO 22196:2011. It also presented the effects of Lapacho on the structural properties and biodegradation rates of biocomposites. This research study provides very important results complementing the current state of knowledge about the biocidal properties of Lapacho from Tabebuia plants and about cortex-reinforced biocomposites.

## 1. Introduction

The demand for polymeric materials of novel properties grows together with both the economic development and the continuous technological progress. A constantly rising amount of plastic waste that loads an natural environment, law regulations, and diminishing resources of petroleum have resulted in increasing interest in biodegradable materials and their modification.

Biodegradable materials are commonly considered as future polymers, mainly because they are plastics undergoing rapid biodegradation under conditions of industrial composting. Because of biocompatibility and biodegradability, they are being applied in medicine and, now to a greater and greater extent, in the packaging industry, pharmaceutical, automotive, biotechnology or horticulture [1,2,3].

The rapid development of specialized applications of polymeric materials in various areas of technology and other aspects of life poses greater and greater qualitative requirements. The common use of these materials implies larger and larger loads of the natural environment, which is increasing because of growing mass of used-up plastic waste. Therefore, the introduction of mass-scale application of biodegradable polymers appears necessary.

The use of polylactid (PLA) in research is reasonable mainly due to its biodegradability. Nowadays, due to an increasing amount of used-up plastic waste, requirements of environmental protection, and obligatory legislation, plastics produced from crude oil are more and more often being replaced by polymers produced from renewable raw materials, such as maize, which decompose into minerals, water, and gases [4,5,6].

PLA has good mechanical, optical, and barrier properties in comparison to conventional polymers based on petroleum. In addition to good properties, PLA has many defects. Low impact resistance, low thermal stability, and high brittleness cause problems of PLA applications [4,7].

Many scientific researches concentrate on the use of natural plants as fillers to produce biodegradable composites. Natural fibers have low environmental influence, and they are renewable, biodegradable, cheap and have low weight. Therefore, biocomposites may be a large range of adoption comparable with synthetic fiber composites. Mainly advantages are as follows: high specific strength and stiffness, ease of separation, biodegradability, and low density [8,9,10,11,12,13].

Therefore, in various industries, such as packaging, automotive, construction, pharmaceutical, biotechnology, or horticulture, there is a growing demand for natural fibers. The rapid growing researches of biocomposites and their growing importance in the industry are reflected by the volume of publications or patents during the recent years [14,15,16,17,18,19,20,21,22].

The latest reports concern the fabrication of newly bionanocomposites. In these researches, there are intensive development and application of nanocomposites containing biopolymers. In recent new articles, the types, preparation methods, properties, and prominent applications of bionanocomposites (e.g., cellulose-based) in various industry sectors are described and discussed [23,24,25,26]. A few works focus on diverse preparation techniques of chitosan-based bionanocomposites and their emerging applications in various sectors [27,28]. Bionanocomposities have been recently developed for the use in food packaging industry. Reference [29] highlights nanofillers which are useful to develop bionanocomposites for food and modify materials of bionanocomposites for food industry.

According to [30], bionanocomposites prepared from cellulose-nanowhisker-reinforced polylactic acid are suitable for packaging materials. A packaging material prepared is biodegradable, with good mechanical properties and thermal stability. Recently developed bionanocomposites have potential to be used as active foods packaging. In [31], the developed bionanocomposite films exhibit strong antioxidant and antimicrobial activity.

A growing interest of natural fibers, which exhibit antimicrobial activity, is observed due to the necessity of sterilizing of various types of equipment (seats, in particular during the pandemic), package trays, pharmaceuticals or medical instruments, manufactured from biodegradable materials [32,33]. Currently, the gamma radiation (high-cost and time-consuming technique) and chemical disinfection methods are commonly used for the sterilization of medical equipment and food packaging. Unfortunately, not all composites are resistant to these factors, and in addition, the used chemicals often change the properties and durability of materials in question. Moreover, residues of disinfectants may have a significant impact on human health. Therefore, the use of natural fibers that exhibit biocidal activities and, at the same time, do not adversely affect the properties of the biocomposites is justified. On the contrary, they may improve the mechanical properties of biocomposites and advantageously affect biodegradation rates.

Wood-fiber-reinforced polymers are used on a large scale in the construction industry, for example in decking. In addition, natural fiber composites are increasingly used in non-construction applications such as surfing board, toys, packaging or etiu for the electronic industry. Due to the wide application of composites for various purposes, it is important to use environmentally friendly materials and modify them with natural plant compounds that show biocidal properties (bactericidal and fungicidal).

Natural fibers can be extracted from different parts of plants or trees, such as the bast, leaf, cortex or wood. Lapachol is a naturally occurring 1,4-naphthoquinone originally isolated by the Italian phytochemist E. Paterno from Tabebuia avellanedae (Bignoniaceae) in 1882 [34]. A wide range of pharmacological activities have been observed for lapachol in the literature, such as anti-inflammatory, antimalarial, antiseptic, antitumor, antiviral, or biocidal (bactericidal, fungicidal, and viricidal) effects [35,36].

The aim of this article is to discuss in detail the physicochemical properties of PLA reinforced by cortex fibers which may cause bacterial mortality and increased biodegradation rates. After the first discovery of Lapachol from Tabebuia avellanedae by E. Paterno and reports on his antiseptic, bactericidal, or fungicidal effects, several reviews have been published since then the use of Lapachol in a wide range of pharmacological activities. However, so far, there have been no research and results available considering Lapachol-reinforced biocomposites. Keeping in mind the importance of biocomposites and increased attention in research communities, one of the key objectives of this study is to show the up-to-date progress in the field of cortex-fibers-reinforced biocomposites with the perspective of applications.

## 2. Materials and Methods

### 2.1. Materials

PLA-type 2003D (Cargill Down LLC, Minnetonka, MN, USA) with a melt flow rate of 4.2 g/10 min. (2.16 kg; at 190 °C) and a density ρ of 1.24 g/cm^3^ was used as a materials matrix.Cortex Lapacho fibers (Posadas, Argentina/Brazil) with a fiber length of 20 mm in the amount of 1–10 wt% were melt-compounded with the PLA matrix.Enzyme Proteinase K from Tritirachium album (Blirt, Poland), a buffer of 0.1 M Tris HCl and sodium azide (NaN_3_), was used for enzymatic degradation tests.

### 2.2. Processing

Samples were prepared on a co-rotating twin-screw extruder-type BTSK 20/40 D (Bűhler, Braunschweig, Germany). Cortex fibers (1–10 wt%) and pure PLA were dried (for 24 h at 60 °C), mixed and added to the extruder feeding zone. The temperatures of the individual zones and head were 180, 182, 184, 186 and 185 °C, respectively. Next, in order to produce biocomposites for mechanical testing, an injection molding machine-type TRX80 ECO60 (Tederic Machinery Co., LDT, Zhejiang, China) was used. The temperatures of zones I, II, and III of the injection molding machine cylinder, head, and mold were 190, 190, 185, 175, and 30 °C, respectively.

The samples are indicated as P and PX, where P indicates pure PLA and PX indicates PLA containing X (*w*/*w*) of the cortex (X = 1, 2, 3, 5, and 10 wt% cortex).

### 2.3. Measurements

Tensile tests were determined on a tensile-testing apparatus according to the standard procedure (PN-EN ISO 527–1). Impact strength tests were determined according to the standard procedure (PN-EN ISO 179-1), using an XJ-5Z (Time Group Inc., Novegro-Tregarezzo, Italy) impact hammer with an about energy of 4 J.

Dynamic mechanical analysis (DMA) measurements were carried out using a Q800 analyzer (TA Instruments, New Castle, DE, USA). DMA measurements were carried out in a dual cantilever mode at a constant frequency of 1 Hz and an amplitude of 15μm as a function of temperature ranging from 25 to 160 °C.

The thermal stability of the biocomposites was evaluated using a thermogravimeter Q500 (TA Instruments, New Castle, DE, USA). The samples of about 19 mg were heated from 20 to 800 °C at a rate of 10 °C/min under nitrogen atmosphere.

Differential scanning calorimetry (DSC) measurements were performed using calorimeter Q200 (TA Instruments, New Castle, DE, USA) under nitrogen flow. The samples was cut off from polymer granules and placed on an aluminum pan for measurements. The samples of about 8 mg were successively quenched to 20 °C, heated to 210 °C at a rate of 10 °C/min, annealed at 210 °C for 3 min, cooled to 20 °C at a rate of 10 °C/min and reheated to 210 °C at a rate of 10 °C/min. In order to eliminate the thermal history of the samples, the measurement results were analyzed based on the data obtained at the second heating.

Images showing changes in the structure of the surface of the samples and those of the cross-sections of biocomposites as well as changes occurring on the surface of the samples subjected to degradation were investigated by using a scanning electron microscope (SEM) Phenom XL (ThermoFisher Scientific, Eindhoven, The Netherlands). The samples were sprayed with a 2 nm layer of gold before testing.

In order to examine the enzymatic degradation, the studied samples of about 15–17 mg were subjected to the action of the selected enzyme. According to this procedure, the samples were placed in test tubes containing of 2 mg Proteinase K, 10 mL of 0.1 M Tris-HCl buffer, 2 mg of NaN_3_. After 24 h, 5 days, and next successive 1 to 8 weeks of incubation, the mass losses of the samples were measured, and their surfaces were analyzed under an SEM. The temperature in the incubator was constant of 37 °C. Each time, the samples withdrawn from the reaction mixture were washed by distilled water and dried in a moisture analyzer MAX 60/NH (Radwag, Radom, Poland), until a constant weight was reached.

The mass loss was calculated using Equation (1):(1)$$\Delta m=\frac{\left(m_{s}\;-\;m_{f}\right)}{m_{s}}\cdot 100\%$$
where *m_s_* is the initial mass (mg) of a sample and m_f_ is the mass (mg) of the sample after the specified period of incubation.

The evaluation of the bactericidal properties of the prepared biocomposites was performed according to the standard ISO 22196:2011. The studies were based on reference bacterial strains, i.e., *Staphylococcus aureus* (ATCC 6538P) and *Escherichia coli* (ATCC 8739). The analysis was carried out in triplicate. A PLA film (without cortex fibers) was as a control sample. Samples in the form of thin films were produced using a compressing holder, attached to the DMA. PLA and biocomposite granules were compressed at 160 °C under a pressure of 15 N.

The control sample and the biocomposites tested were covered with the suspensions of bacterial strains investigated with a specified number of cells. All tested samples were left for a specified period of time (0 and 24 h). After this time, bacterial cells were recovered from the surface and suspended in a solution containing soybean casein digest broth with leticin as a neutralizer. Then, the number of cells which were living and able to grow was investigated by the inoculation on Plate Count Agar in triplicate. The plates were incubated at 37 °C for 24 h.

The reduction (R) of the number of living cells of the tested bacteria was calculated using Equation (2):R = (U_t_ − U_0_) − (W − U_0_),(2)
where U_0_ is the average of the logarithm of the number of viable bacteria recovered from the control samples immediately after inoculation (validation of recovery efficiency); U_t_ is the average of the logarithm of the number of viable bacteria recovered from the control samples after 24 h (controls of survival in time); and W is the average of the logarithm of the number of viable bacteria recovered from the test samples after 24 h.

According to the standard ISO 22196:2011, the reductions of the numbers of bacterial cells which were living and able to grow (e.g., *Escherichia coli* and *Staphylococcus aureus*) by at least two orders of magnitude were considered as biocidal effects. Therefore, R of ≥2 was interpreted as a bactericidal effect of the investigated biocomposites.

## 3. Results

### 3.1. Mechanical Properties

The results of the mechanical properties of the PLA and the biocomposites are presented in Table 1.

The results of tensile strength and modulus as a function of cortex content are presented in Figure 1 and Figure 2. It has been shown that the increased cortex fibers content decreased the biocomposites strength while the modulus was slightly increased with the increased cortex fibers content. The addition of cellulose contains in cortex increased the modulus of the biocomposites, while the strength of the biopcomposites decreased significantly.

The results of the impact strength of the biocomposites at various cortex fibers contents are presented in Figure 3. The impact strength of the biocomposites decreased with the increase of the cortex fibers content. The results showed that the addition of natural fibers was ineffective to improve the brittleness of the biocomposites. The decrease of the impact strength of the biocomposites was effected from the poor interfacial adhesion between the cortex fibers and the matrix, as is shown by SEM images.The interfacial adhesion strength between the cortex fibers and the matrix affected the impact property of the biocomposites, because the impact energy was dissipated by debonding, cortex, and/or matrix breakage and the cortex fibers were pulled outwards.

The results of the storage moduli of the tested samples are presented in Figure 4, while the results of the damping coefficients (tanδ) are shown in Figure 5. For the sake of the discussion clarity, only the results for samples P, P5, and P10 are presented.

As seen from Figure 4, four characteristic temperature ranges related to PLA phase states can be distinguished: (i) glassy state to about 60 °C, (ii) glass transition from about 60 to about 90 °C, (iii) rubber plateau around 90 °C, and (iv) cold crystallization (from about 95 to about 150 °C). In the glassy state, a small content of cortex (up to 5 wt%) reduced the storage modulus from 2776 MPa (sample P) to 2290 MPa (sample P5), while the 10% cortex content increased the storage modulus to about 3482 MPa (sample P10). It is possible that there are two competing effects: the plasticizing effect related to the extraction of oils and other plant substances during composite processing and the other associated with the mechanical reinforcement effect typical for cellulose fillers. Only when the filler content was significant (10 wt%), the second reinforcement effect was predominant.

In the glass transition temperature range, no significant changes were detected (Figure 5). The maximum values of tanδ were around 71 °C (glass transition temperature). The lower damping coefficient for sample P10 can reflect a significantly lower share of PLA (90%) in this composite.

In the cold crystallization temperature range, increases in the storage moduli of samples containing more cortex contents can be seen at lower temperatures and with more sharp characteristic increases. This can demonstrate the susceptibility of this filler as a nucleating agent for PLA crystallites, which, during formation, caused a significant increase in the storage modulus.

### 3.2. Material Structure

The thermal properties of the tested samples, determined by DSC tests, are presented in Table 2.

The T_g_ of sample P was 60.2 °C. The introduction of cortex into the PLA matrix caused a slight decrease in the glass transition temperature of the obtained materials. The recorded T_g_ values ranged from 59.9 °C for sample P1 to 58.8 °C for sample P10. Thus, it can be seen that the cortex or substances contained in it had low plasticizing properties of PLA, and the plasticizing effect increased with the increase in the filler content.

The plasticizing properties of the cortex were also visible in the course of the cold crystallization process (Figure 6). It can be seen from Figure 6 that fibers acted as nucleants for the cold crystallization of PLA. This effect was also reported elsewhere [37].

The T_cc_ value for sample P was 127.1 °C. The intensity of the cold crystallization process was low, which resulted into a small ΔH_cc_ value of 4.4 J/g. The introduction of cortex to the polymer matrix resulted in a significant decrease in T_cc_ and a significant increase in the intensity of the cold crystallization process, which resulted in much higher values of ΔH_cc_. The T_cc_ value of sample P1 decreased to 122.8 °C, while the ΔH_cc_ value increased to 20.4 J/g. The greatest changes were observed with the highest cortex content in the material. The T_cc_ value of sample P10 further decreased to 118.2 °C, and the ΔH_cc_ value increased to 26.9 J/g.

The increase in the intensity of the cold crystallization process translated into an increase in the intensity of the melting process, in which the crystalline phase formed in the cold crystallization process was melted. Thus, similar to the increase in the value of ΔH_cc_, the values of ΔH_m_ also increased. For sample P, the ΔH_m_ value was 5.9 J/g and increased significantly after the addition of cortex to 21.0 J/g for sample P1 and to 28.1 J/g for sample P10.

The changes in the intensity of the melting process were not accompanied by the changes in the melting point of the crystalline phase. Tm values of all the tested materials ranged from 148.3 to 149.6 °C. The lack of major changes proved that the crystal structures formed in the materials during cold crystallization were very similar and the content of the bark and its amount in the material did not significantly affect the structures.

Regardless of the course of cold crystallization and melting processes, the degrees of crystallinity (X_c_) of all the tested samples were practically the same. The X_c_ values for all samples ranged from 0.6% to 1.3%. This was due to the similar values of ΔH_cc_ and ΔH_m_, which means that the entire melting crystal phase was formed in the process of cold crystallization and was not present in the starting material. On the basis of the obtained values, it can be concluded that the tested materials were practically completely amorphous, regardless of the cortex content.

Cortex is a natural fiber that mainly consists of cellulose, hemicellulose, lignin, and extractives. These components are responsible for the chemical and physical properties of cortex as well as those of biocomposites. Because cellulose, hemicellulose, or lignin degrades at different temperatures and different rates, they may have a negative influence on the mechanical properties of produced biocomposites [38,39].

A very important property of cortex fibers is their thermal stability (Figure 7). Before the experiment, cortex fibers were dried at 60 °C for 24 h. Despite drying, about 5 wt% of water confined in cortex fibers was evaporated when heated to 100 °C. The thermal degradation of biocomposites is a very important aspect for the production process. Due to its lower thermal stability, natural fibers are usually used as fillers only in materials that are processed at temperatures below 200 °C. Thermal degradation determines the maximum processing temperature that can be used [39]. It was found that fibers are thermally stable at about 200 °C, and for that reason, the sample processing temperature was set below this value. When using natural fibers as reinforcement in thermoplastics, one has to keep in mind that temperature during extrusion and injection molding can be a crucial factor affecting the mechanical properties of the composites. The degradation of natural fibers due to high temperatures can lead to undesirable effects, such as decoloration or a decrease in mechanical strength [40,41].

A small weight loss was observed at 100 °C, probably related to water evaporation. The thermal degradation rate gradually increased above 200 °C, and then, as heating continues, the weight loss rate increased. There are two main stages of cortex degradation: below 350 °C and at higher temperatures between 350 and 500 °C. The first stage can be attributed to the degradation of individual cortex–wood components such as hemicelluloses. The second degradation stage can be attributed to the decomposition of cellulosic and lignin materials in the cortex [42,43].

The peaks at 280 and 400 °C were wide, which may means there was overlapping between the decomposition of hemicelluloses, cellulose, lignin, and extractives in cortex. The decomposition reactions may occur simultaneously. According to [44], in some cortex–wood species, the decomposition of cortex can continue at temperatures up to 550 °C, overlapping with the decomposition of polysaccharides and lignin.

According to [45], some wood components start to decompose at around 175 °C, and low temperature degradation at a low rate occurs in lignin and hemicelluloses. In [46], the degradation of cellulose takes place at high temperature associated with the pyrolitic degradation of lignin. In [47], it was confirmed that hemicelluloses degrade between 180 and 350 °C, the degradation of lignin takes place between 250 and 500 °C, and cellulose degrades between 275 and 350 °C [39].

The different behaviors observed possibly result from the differences in the structures and chemical properties of the cortex components (e.g., oils extracts fraction). Hemicellulose is least thermally stable than cellulose. Hemicellulose consists of saccharides, mainly xylan (least thermally stable) and glucomannan (in different amounts). Moreover, hemicellulose has an amorphous structure and is rich in branches, which easily degrades at low temperatures to CO, CO_2_, and hydrocarbon. Hemicellulose evolves more noncombustible gases and less tar than cellulose, whereas cellulose has an ordered structure and consists of a long polymer of glucose without branches. Different from hemicellulose, cellulose is principally responsible for the production of flammable volatiles. The degradation of lignin takes place at temperature between 225 and 450 °C. Such a wide degradation temperature range is associated with the activity of chemical bonds. Lignin has a lot of aromatic rings with various branches. The break of aryl-alkyl-ether linkages occurs between 150 and 300 °C. At a temperature of about 300 °C, aliphatic side chains split off from the aromatic ring. At 370–400 °C, the carbon–carbon linkage between lignin structural units is cleaved. During the degradation of lignin, more residual char is produced than during the degradation of cellulose [48,49,50].

The thermal decompositions of the PLA and the biocomposites are illustrated in Figure 8. The temperatures of decomposition (T_d_) and temperatures for mass losses of 5%, 25%, 50%, and 95% (T_5%_, T_25%_, T_50%_, and T_95%_, respectively), derived from thermogravimetry (TG) and TG/DT curves, as well as the remaining mass at 450 °C, are summarized in Table 3.

The thermal degradation of biocomposites started from about 200 °C, causing a rapid weight loss at about 380 °C. This stage can be attributed to the degradation of hemicellulose and cellulose [16,18]. PLA was thermally stable at temperatures up to 300 °C and then began to degrade rapidly in a single stage between 300 and 380 °C.

In Table 3, the temperatures at which the 5%, 25%, 50%, and 95% of weight were lost are listed for all biocomposites. It can be seen that the addition of 1 and 2 wt% cortex fibers resulted in an increase in the thermal stability of the composites, with increases from about 13 and 9 °C, respectively. However, the further addition of cortex fibers caused a reduction in the thermal stability of the biocomposites, with a decrease by 4 °C when the PLA matrix was reinforced with a 5 wt% of cortex.

A rise in the degradation temperature for biocomposites can be attributed to higher cellulose and lignin content in the samples and the influence of oils and other substances occurring in the cortex of Lapacho. The oils and other substances occurring in the cortex of Lapacho presumably increased the durability of the composites, which led to the rise in the temperature, at which their thermal decomposition started.

### 3.3. Enzymatic Degradation Testing

Figure 9 shows the influence of enzyme Proteinase K on the rate of the degradation of pure PLA and PLA with 1%, 5%, and 10% cortex fibers. The mass loss in PLA ranged from 0.2% to 2.2%, 0.3% to 2.6%, and 0.3% to 3.8%, and for sample P10, the mass loss in PLA ranged from 0.4% after 1 day to 5.8% after 28 days.

The weight loss in biocomposites is mainly caused by the cortex fibers conent. With the increase in the cortex fibers content in the samples, the weight loss of the biocomposites increased (compared with that of PLA). It is possible that the increased weight reduction of biocomposites (compared with that of PLA) is caused by two effects, i.e., direct and indirect. The first is related to the degrading cortex fibers, and the second effect is related to the growing contact surface of PLA. This may be explained by the fact that cortex fibers acted like a transport medium for water and nutrients. Pinholes, cracks, and cavities, which are present in the biocomposites after the degradation process (see SEM images), lead to the fact that water and nutrients can easily penetrate into the cortex and the biocomposites. Additionally, the ingress of nutrients can stimulate the rapid development of bacteria or fungi. Such microbiological colonization increases the effectiveness of enzyme activity and leads to the faster degradation of the biocomposites [51].

### 3.4. Surface Morphology Changes

The SEM images of the pure PLA and the biocomposites before and after treatment by proteinase K are shown in Figure 10, Figure 11, Figure 12, which were taken after the end of biodegradation. Degradation appeared clearly on the surface.

Pinholes, cracks and cavities apperaed in the biocomposites during enzyme action and increased as the duration of the degradation proces was increased. Thereby, the structure changes in the biocomposites became larger and larger. Additionally, as the cortex fibers content was increased, the changes in the biocomposites increased. The analysis of the SEM images confirmed the results of the sample mass losses.

The cortex fibers have the natural spaces, which water, bacteria, or fungi can easy inflintrate into the PLA matrix from the inside of the sample. The holes and spaces in the biocomposites grew during the degradation process. It may be attributed to the hydrolysis proces of the biocomposites. In addition, cellulose, lignin, pectin, and tannins, which are the components of the cortex fibers used, increased the efficiency of enzymes and thus influenced the rate of the sample degradation.

### 3.5. Antimicrobial Properties

The reductions (R) of the bacterial strains of *Escherichia coli* and *Staphylococcus aureus* for the biocomposites are listed in Table 4.

According to the standard ISO 22196:2011, a biocidal agent may be considered as effective, if the R value resulting from the agent’s action is at least 2. The results shown in Table 4 indicated that the Lapacho cortex was not an active biocide against *Escherichia coli* and *Staphylococcus aureus* bacterial strains.

In both cases, the reductions of the numbers of living and viable cells of the tested bacteria were only 1, which means that the biocidal activity of oils and other substances occurring in the cortex of Lapacho was not effective (for the produced biocomposites). The test results did not confirm wide-range activities for Lapachol which are presented in the literature, such as bactericidal effects [34,35,36].

In the case of the *Escherichia coli* bacterial cells, the R value was 1 only for biocomposites with a 10 wt% cortex, whereas in the case of the *Staphylococcus aureus* bacterial cells, the R value was 1 for the biocomposites with a cortex content of 2 wt%. Probably, *Staphylococcus aureus* bacterial cells were more susceptible to the action of the individual components of the oils contained in the Lapacho cortex; therefore, their mortality was higher at lower concentrations. However, the results still not meet the requirements of the standard ISO 22196:2011 and does not allow considering Lapacho cortex (in the range of the content (wt%) applied) to be a biocidal factor.

## 4. Conclusions

This study was carried out to investigate the use of cortex fibers as reinforcement in a PLA matrix and the production of biocomposites which may cause bacterial mortality and increase biodegradation rate.

The evaluation of the mechanical properties of the composites demonstrated that the tensile modulus was improved with the increased cortex fibers content but tensile strength and elongation to break were decreased. The addition of natural fibers reduced the impact strength of PLA composites. Poor interfacial bonding resulted in poor energy dissipation between the natural fibers and the PLA matrix.

The addition of cortex fibers caused two competing effects: the plasticizing effect related to the extraction of oils and other plant substances occurring in Lapacho cortex during the composite processing and the other one associated with the mechanical reinforcement effect typical for cellulose fillers. However, the reinforcement effect was observed, only when the filler content was significant.

Thermal stability increased with the addition of cortex fibers reinforcement. The oils and other substances occurring in the cortex of Lapacho presumably increased the durability of the composites, which led to a rise in the temperature at which their thermal decomposition started.

The biodegradation studies on the samples showed that the biocomposites disclosed a higher weight loss compared with pure PLA. The cortex fibers acted like a transport medium for water or nutrients and can easily infiltrate into the cortex and reach at the PLA matrix from the inside of the sample. It is very important and necessary for hydrolytic weathering. Moreover, cortex fibers supported bacteria/fungi colonization, stimulating extensive microbiological invasion and colonization. In addition, cellusose which is the main structural compound of cortex, increased the effectiveness of enzymatic activity.

The results shown indicated that the Lapacho cortex is not an active biocide against the *Escherichia coli* and *Staphylococcus aureus* bacterial strains. The produced biocomposites caused bacterial mortality but still do not meet the requirements of the standard ISO 22196:2011. In all biocomposites, the reductions of the numbers of living and viable cells of the tested bacteria were only 1, which means that the biocidal activities of oils and other substances occurring in the cortex of Lapacho were not effective.

## Figures and Tables

**Figure 1 materials-14-04241-f001:**
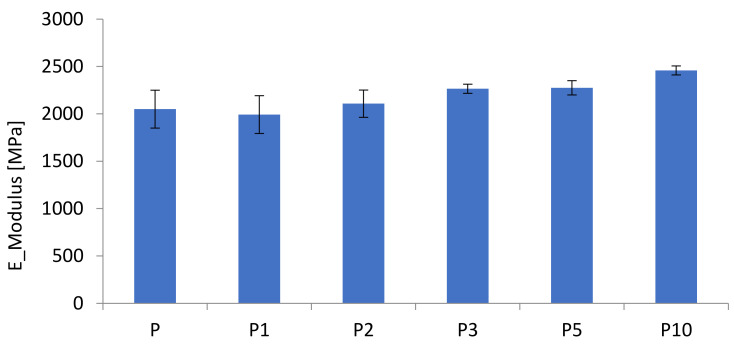
Tensile strengths of the biocomposites.

**Figure 2 materials-14-04241-f002:**
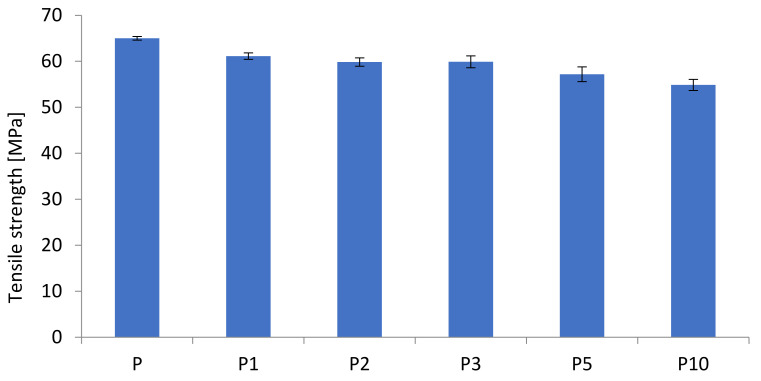
E-moduli of the biocomposites.

**Figure 3 materials-14-04241-f003:**
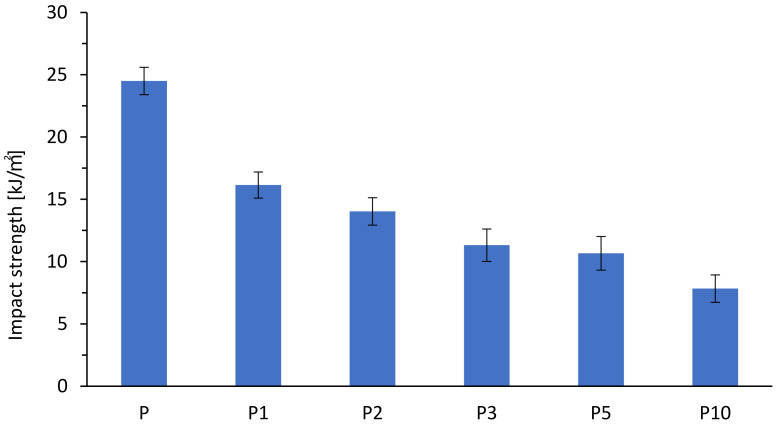
Impact strengths of the biocomposites.

**Figure 4 materials-14-04241-f004:**
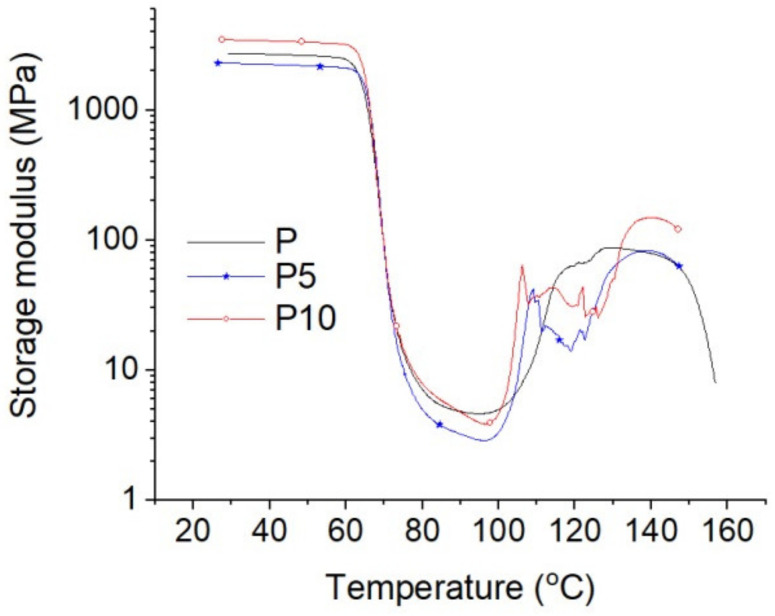
Storage moduli of samples P, P5, and P10.

**Figure 5 materials-14-04241-f005:**
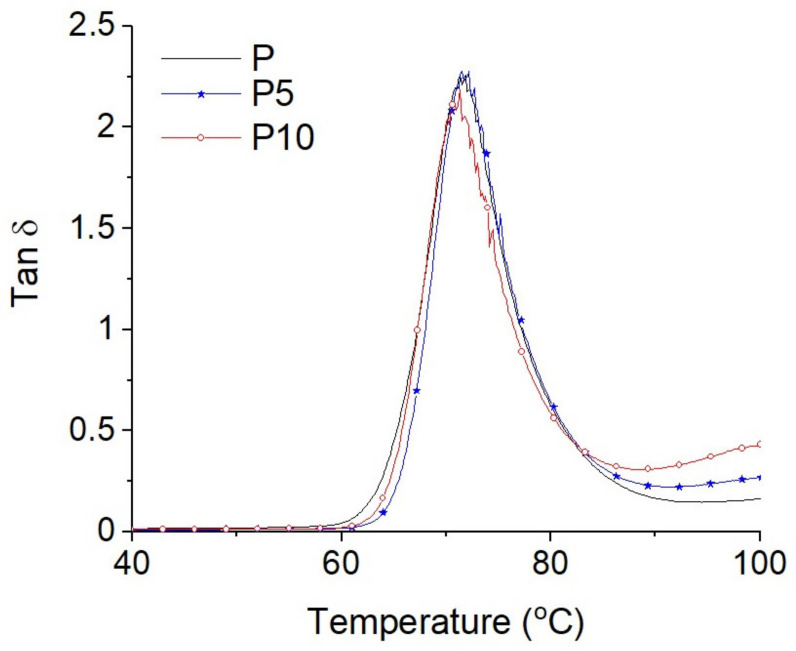
Damping coefficients (tanδ) of samples P, P5, and P10.

**Figure 6 materials-14-04241-f006:**
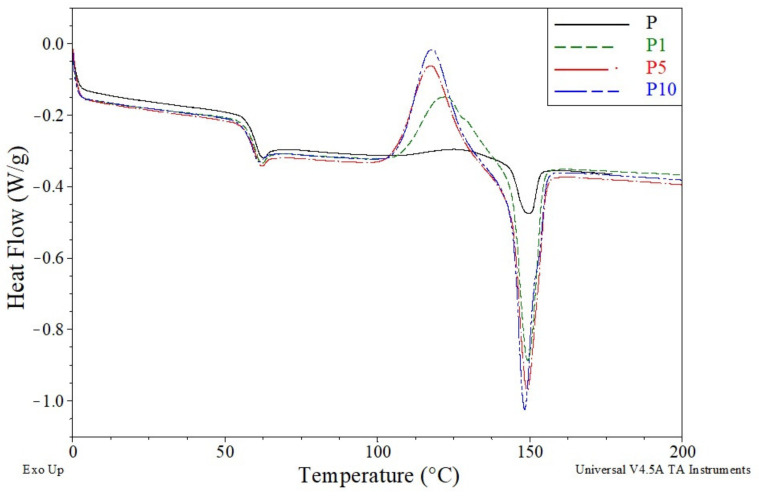
Differential scanning calorimetry (DSC) thermograms of samples P, P1, P5, and P10.

**Figure 7 materials-14-04241-f007:**
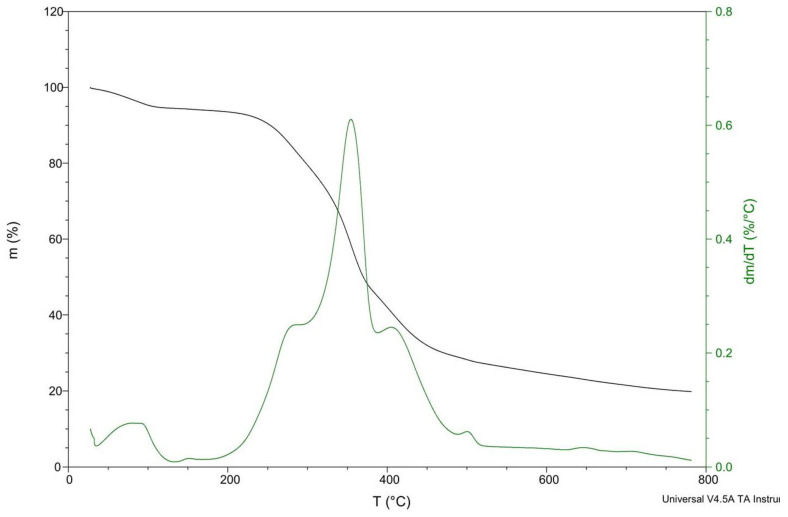
Thermogravimetric curves for cortex fibers.

**Figure 8 materials-14-04241-f008:**
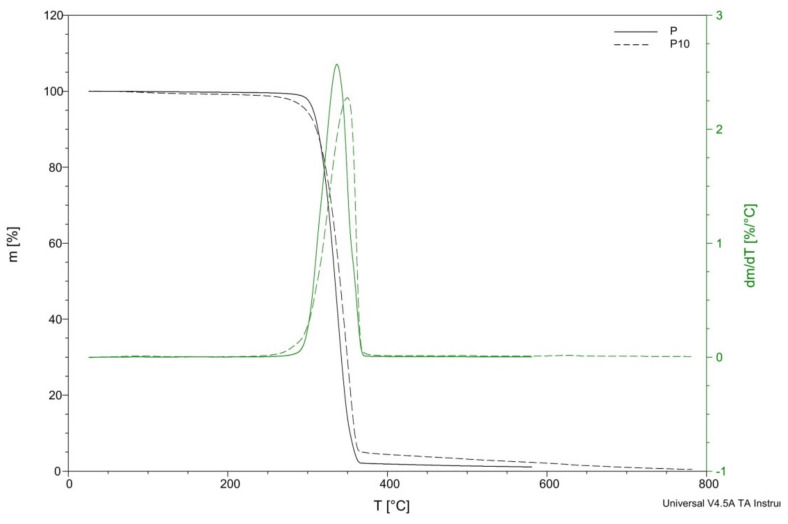
Thermogravimetry (TG) curves for samples P and P10.

**Figure 9 materials-14-04241-f009:**
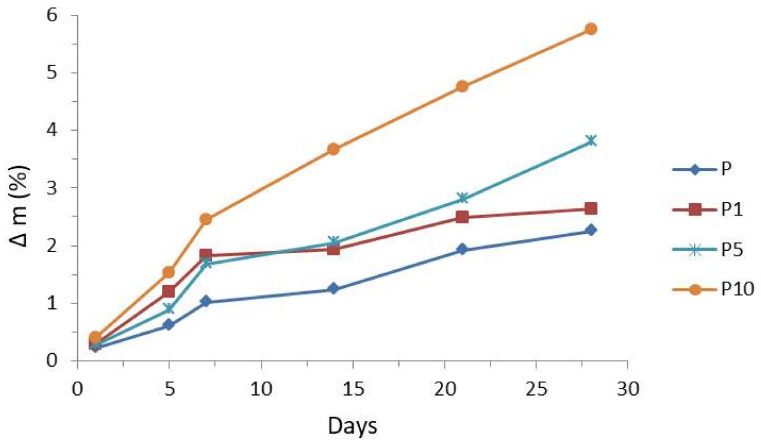
Effects of proteinase K on the degradation rates of samples P, P1, P5, and P10.

**Figure 10 materials-14-04241-f010:**
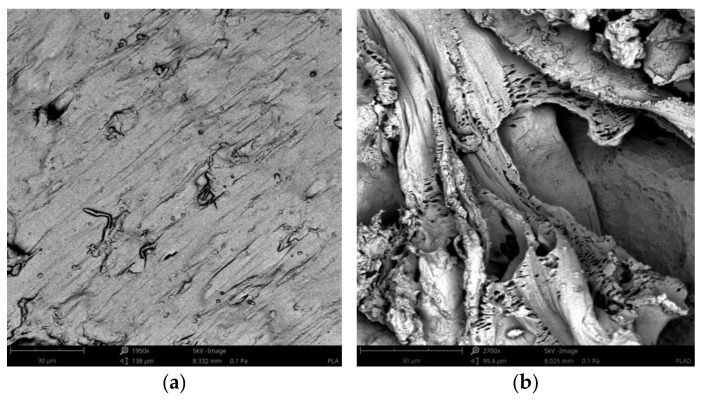
SEM images of sample P before the treatment by proteinase K (**a**) and sample P after the treatment by proteinase K (**b**).

**Figure 11 materials-14-04241-f011:**
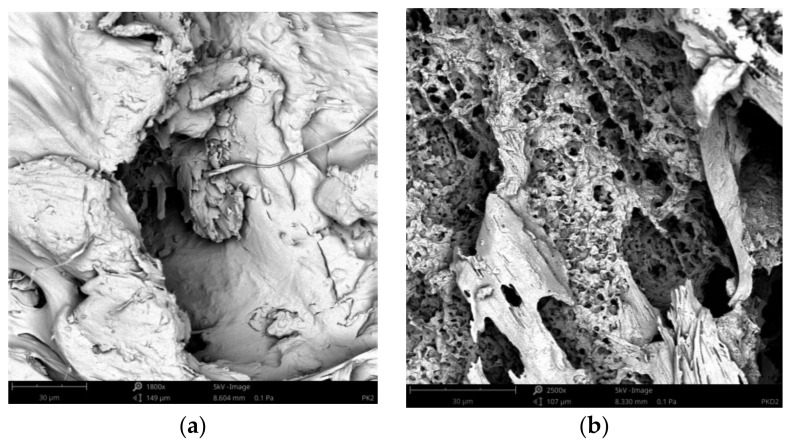
SEM images of sample P2 before the treatment by proteinase K (**a**) and sample P2 after the treatment by proteinase K (**b**).

**Figure 12 materials-14-04241-f012:**
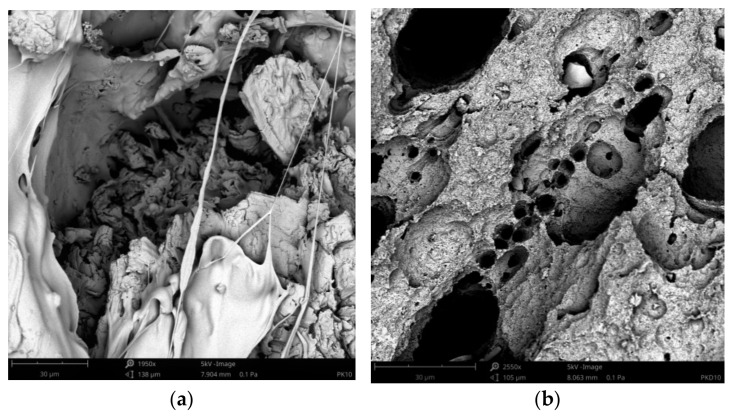
SEM images of sample P10 before the treatment by proteinase K (**a**) and sample P10 after the treatment by proteinase K (**b**).

**Table 1 materials-14-04241-t001:** Mechanical properties polylactide (PLA) and biocomposites.

Sample	Tensile Strength (MPa)	Elongation at Break (%)	E-Modulus (MPa)
P	65.0 ± 0.4	4.58 ± 0.17	2050 ± 200
P1	61.1 ± 0.7	3.89 ± 0.23	1992.58 ± 190
P2	59.8 ± 0.9	3.72 ± 0.15	2107.93 ± 144
P3	59.9 ± 1.3	3.51 ± 0.15	2264.97 ± 48
P5	57.2 ± 1.6	3.19 ± 0.32	2275.01 ± 75
P10	54.9 ± 1.2	2.87 ± 0.24	2458.80 ± 47

**Table 2 materials-14-04241-t002:** Thermal properties of the tested samples.

Sample	T_g_ (°C)	T_cc_ (°C)	ΔH_cc_ (J/g)	T_m_ (°C)	ΔH_m_ (J/g)
P	60.2	127.1	4.4	149.6	5.2
P1	59.8	122.8	20.4	149.3	21.0
P2	59.9	121.2	22.4	148.9	23.1
P3	59.8	119.6	23.1	148.6	23.8
P5	59.8	117.9	26.7	149.2	27.5
P10	58.8	118.2	26.9	148.3	28.1

**Table 3 materials-14-04241-t003:** Temperatures of decomposition (T_d_) and temperatures (T_5%_, T_25%_, T_50%_, and T_95%_) corresponding to mass losses of 5%, 10%, and 95%, derived from TG and TG/DT curves.

Sample	T_d_ (°C)	T_d/dt_ (°C)	T_5%_ (°C)	T_25%_ (°C)	T_50%_ (°C)	T_95%_ (°C)	Residue at 450 °C (%)
P	315.68	337.32	306.94	323.25	334.36	358.78	1.6
P1	328.99	356.62	312.45	335.69	348.08	367.51	1.36
P2	325.14	355.62	306.28	330.95	344.16	363.43	1.42
P3	316.48	351.86	301.63	324.66	338.35	359.97	1.66
P5	311.91	344.86	284.07	317.34	333.84	357.53	3.2
P10	321.74	349.71	298.30	326.45	340.63	368.96	3.8

**Table 4 materials-14-04241-t004:** The reductions of the numbers of living and viable cells of the tested bacteria.

Tested Bacteria *	Cortex Content (%)									
	1		2		3		5		10	
	×10^6^ (jtk/mL)	R	×10^6^ (jtk/mL)	R	×10^6^ (jtk/mL)	R	×10^6^ (jtk/mL)	R	×10^6^ (jtk/mL)	R
*Escherichia coli* (*E. coli*)	1.1	0	1.2	0	1.0	0	1.0	0	0.3	1
*Staphylococcus aureus* (*S. aureus*)	1.0	0	0.9	1	0.9	1	0.9	1	0.1	1

* The numbers of cells of the tested strains in the prepared suspension for testing: *E. coli*, 1.4 × 10^6^; *S. aureus*, 1.6 × 10^6^. The time of the contact of bacteria with the tested foil was 24 h.

## Data Availability

Data are contained within the article.

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
