# Peer review of "Evaluation of the Mechanical and Biocidal Properties of Lapacho from Tabebuia Plant as a Biocomposite Material"

_materials, 2021, doi:10.3390/ma14154241_

Round 1
Reviewer 1 Report
The paper “Evaluation of the mechanical and biocidal properties of Lapacho from Tabebuia plant as a biocomposite material”, by Magdalena StepczyĹ„ska M, Alona PawĹ‚owska, Krzysztof Moraczewski, Piotr Rytlewski, Andrzej Trafarski, Daria Olkiewicz, Maciej Walczak, presents studies related to physicochemical properties polylactide (PLA) reinforced by cortex fibers. The paper is well written, and recommended to be accepted after minor corrections.
- General Remark: the authors are kindly asked to choose the writing form of the ISO standard. In the manuscript appears 3 forms: “ISO 22196:2011”, “ISO 22196, 2011” and “ISO 22196”. Please choose 1 form, and use it inside the manuscript.
- Line 11: please correct the word “fibres”, with the correct English form “fibers”.
- Lines 18-20 from the abstract, have quite the same form like at the end of the Introduction section, mainly lines 93-94. The authors are asked to reformulate in the abstract, or in the Introduction those lines.
- Line 112: please use superscript for “150 min-1”.
- Lines 116-117: please insert the degree superscript symbol (is missing) for “175C, and 30C”.
- Lines 182 – 188: the authors have used the expression “… the common logarithm…”. What should denote this expression, because mathematically speaking, you don’t have nothing “common” in the basic mathematics functions. Indeed, you have the logarithmic function, but you will not find it written as “the common logarithmic function”. As far as I know, in mathematics domain exist only 1 logarithmic function, known as the inverses of exponential function. Please reformulate the expression (remove the word “common”).
- Figure 11: The SEM from “a) P2” it’s at a different magnitude/ scale, than the rest of the SEM photos. Please check again to be sure that all the images are at the same magnitude, in order to see precisely the differences between conditions.
- Line 457: the ISO standard has a missing number “ISO 22196:201”. Please make corrections.
Author Response
Thank you very much for your reviews. After a profound analysis of the review sent, I am herewith sending the comments to the review.
- General Remark: the authors are kindly asked to choose the writing form of the ISO standard. In the manuscript appears 3 forms: “ISO 22196:2011”, “ISO 22196, 2011” and “ISO 22196”. Please choose 1 form, and use it inside the manuscript.
Comment: The suggestions of the Reviewer have been considered. I choosed ISO 22196:2011 form.
- Line 11: please correct the word “fibres”, with the correct English form “fibers”.
Comment: The suggestions of the Reviewer have been considered. Also improved further in the text (in addition to the titles of literature).
- Lines 18-20 from the abstract, have quite the same form like at the end of the Introduction section, mainly lines 93-94. The authors are asked to reformulate in the abstract, or in the Introduction those lines.
Comment: The suggestions of the Reviewer have been considered. I deleted the repetition.
- Line 112: please use superscript for “150 min-1”.
Comment: The suggestions of the Reviewer have been considered.
- Lines 116-117: please insert the degree superscript symbol (is missing) for “175C, and 30C”.
Comment: The suggestions of the Reviewer have been considered.
- Lines 182 – 188: the authors have used the expression “… the common logarithm…”. What should denote this expression, because mathematically speaking, you don’t have nothing “common” in the basic mathematics functions. Indeed, you have the logarithmic function, but you will not find it written as “the common logarithmic function”. As far as I know, in mathematics domain exist only 1 logarithmic function, known as the inverses of exponential function. Please reformulate the expression (remove the word “common”).
Comment: The suggestions of the Reviewer have been considered.
- Figure 11: The SEM from “a) P2” it’s at a different magnitude/ scale, than the rest of the SEM photos. Please check again to be sure that all the images are at the same magnitude, in order to see precisely the differences between conditions.
Comment: The suggestions of the Reviewer have been considered.
- Line 457: the ISO standard has a missing number “ISO 22196:201”. Please make corrections.
Comment: The suggestions of the Reviewer have been considered.
Reviewer 2 Report
Recommendation:
This is review for ‘Evaluation of the mechanical and biocidal properties of Lapacho from Tabebuia plant as a biocomposite material’ by Magdalena StepczyĹ„ska M et al. This work was highlighting the effects of Lapacho fibers, which can be extracted from different parts of the plant or tree, on the mechanical and biocidal properties of the biocomposites. Moreover, the discussion about the physicochemical properties polylactide (PLA) reinforced by cortex fibres, which may cause bacterial mortality and positive on increased biodegradation rate, could be strengthened. Overall, the experimental method descriptions were detailed enough, and the analysis and scientific discussion about biocidal properties of Lapacho from Tabebuia plant and about cortex reinforced biocomposites is persuasive. This paper is interesting, written nicely and related experiments was done carefully. I recommend ‘minor revision’. Specific comments given below should be addressed during revision.
- What are the findings of this research? What is the relevance and novelty of this study? Please add these information in the Introduction part.
- Newly developed bionanocomposites related recent reports have not been discussed much. Authors need to add few important references highlighting recently developed bionanocomposites with more details and references.
- There are no error bars in Fig.1 through Fig.3. Please add error bars.
- Data curves for P, P1, P5, and P10 have all same line style in Fig. 6.
Please use different styles of line for each data curve. For example, P1 for dashed line and P5 for dotted line. - Please highlight which curve represent m(%) or dm/dT (%C) in Fig 7. and Fig. 8.
- It is hard to see the scale bars in Fig. 11 and Fig. 12. Please make them visible.
Author Response
After a profound analysis of the review sent, I am herewith sending the comments to the review.
- What are the findings of this research? What is the relevance and novelty of this study? Please add these information in the Introduction part.
Comment: The suggestions of the Reviewer have been considered.
- Newly developed bionanocomposites related recent reports have not been discussed much. Authors need to add few important references highlighting recently developed bionanocomposites with more details and references.
Comment: The suggestions of the Reviewer have been considered.
- There are no error bars in Fig.1 through Fig.3. Please add error bars.
Comment: The suggestions of the Reviewer have been considered.
- Data curves for P, P1, P5, and P10 have all same line style in Fig. 6. Please use different styles of line for each data curve. For example, P1 for dashed line and P5 for dotted line.
Comment: The suggestions of the Reviewer have been considered.
- Please highlight which curve represent m(%) or dm/dT (%C) in Fig 7. and Fig. 8.
Comment: The suggestions of the Reviewer have been considered.
- It is hard to see the scale bars in Fig. 11 and Fig. 12. Please make them visible.
Comment: I sent pictures in seperates files. In original size are visible.